# In Vitro Activity of Cefiderocol against Clinical Gram-Negative Isolates Originating from Germany in 2016/17

**DOI:** 10.3390/antibiotics12050864

**Published:** 2023-05-06

**Authors:** Esther Wohlfarth, Michael Kresken, Fabian Deuchert, Sören G. Gatermann, Yvonne Pfeifer, Niels Pfennigwerth, Harald Seifert, Paul G. Higgins, Guido Werner

**Affiliations:** 1Antiinfectives Intelligence GmbH, c/o Rechtsrheinisches Technologie- und Gründerzentrum, Gottfried-Hagen-Straße 60-62, 51105 Cologne, Germany; 2German National Reference Centre for Multidrug-Resistant Gram-Negative Bacteria, Departement of Medical Microbiology, Ruhr-University Bochum, 44801 Bochum, Germany; 3Division 13 Nosocomial Pathogens and Antibiotic Resistances, Department of Infectious Diseases, Robert Koch Institute, Burgstraße 37, 38855 Wernigerode, Germany; 4Institute for Medical Microbiology, Immunology and Hygiene, Faculty of Medicine and University Hospital Cologne, University of Cologne, 50935 Cologne, Germany; 5German Centre for Infection Research, Partner Site Cologne-Bonn, 50935 Cologne, Germany; 6Center for Molecular Medicine Cologne, Faculty of Medicine and University Hospital Cologne, University of Cologne, 50935 Cologne, Germany

**Keywords:** Enterobacterales, *Acinetobacter baumannii*, *Stenotrophomonas maltophilia*, *Pseudomonas aeruginosa*, carbapenemase, ESBL, cefiderocol

## Abstract

Antimicrobial resistance poses a global threat to public health. Of great concern are *Acinetobacter baumannii*, *Pseudomonas aeruginosa* and Enterobacterales with resistance to carbapenems or third-generation cephalosporins. The aim of the present study was to investigate the in vitro activity of the novel siderophore cephaloporin cefiderocol (CID) and four comparator β-lactam-β-lactamase-inhibitor combinations and to give insights into the genetic background of CID-resistant isolates. In total, 301 clinical Enterobacterales and non-fermenting bacterial isolates were selected for this study, including randomly chosen isolates (set I, n = 195) and challenge isolates (set II, n = 106; enriched with ESBL and carbapenemase producers, as well as colistin-resistant isolates). Isolates displayed CID MIC_50/90_ values of 0.12/0.5 mg/L (set I) and 0.5/1 mg/L (set II). Overall, the CID activity was superior to the comparators against *A. baumannii*, *Stenotrophomonas maltophilia* and set II isolates of *P. aeruginosa*. There were eight CID-resistant isolates detected (MIC > 2 mg/L): *A. baumannii* (n = 1), *E. cloacae* complex (n = 5) and *P. aeruginosa* (n = 2). Sequencing analyses of these isolates detected the acquired β-lactamase (*bla*) genes *bla*_NDM-1,_ *bla*_SHV-12_ and naturally occurring *bla*_OXA-396_, *bla*_ACT-_type and *bla*_CMH-3_. In conclusion, CID revealed potent activity against clinically relevant organisms of multidrug-resistant Enterobacterales and non-fermenters.

## 1. Introduction

The emergence of antibiotic-resistant bacteria has been described as one of the biggest threats to global health and food safety [1,2]. It is a consequence of selective pressure caused by a range of factors such as the overuse of antibiotics in human and veterinary medicine, as well as insufficient hygiene precautions and the release of antibiotics into the environment [1,2,3,4]. The three most critical pathogens defined by the World Health Organization (WHO) for finding new treatment options are the Gram-negative pathogens *Acinetobacter baumannii* (carbapenem-resistant), *Pseudomonas aeruginosa* (carbapenem-resistant) and the order of Enterobacterales (carbapenem-resistant and ESBL-producing) [5]. Various established antibiotics have been suggested as an appropriate therapy for severe infections caused by carbapenem-resistant Gram-negative bacteria, such as high dose meropenem and colistin [6,7]. New antimicrobial agents, especially β-lactam–β-lactamase-inhibitor combinations such as ceftazidime–avibactam, ceftolozane–tazobactam, imipenem–relebactam or meropenem–vaborbactam, have been considered for the effective treatment of severe infections caused by carbapenem-resistant Gram-negative bacteria [8,9]. However, none of these agents address all resistant Gram-negative bacteria or show sufficient activity against *Stenotrophomonas maltophilia* [7,8,9]. 

Cefiderocol (CID) is a novel parenteral cephalosporin carrying a catechol moiety at the 3-position side chain [10]. It has been shown recently that the compound forms a chelating complex with extracellular trivalent iron, leading to active transport of the drug into the periplasmatic space of *P. aeruginosa* [11]. It is thus considered a siderophore cephalosporin. CID shows promising antimicrobial activity against critical Gram-negative pathogens, such as carbapenem-resistant non-fermenters such as *A. baumannii* (CRAB), *P. aeruginosa* and *S. maltophilia*, as well as carbapenem-resistant Enterobacterales [12], with high stability against β-lactamases of all Ambler classes. Clinically relevant carbapenem-hydrolyzing β-lactamases, such as class A KPC, class B metallo-enzymes or class D OXA enzymes (e.g., OXA-48 in *Klebsiella pneumoniae* or OXA-23 in *A. baumannii*), show the weak hydrolysis of CID [13]. Furthermore, CID shows a low tendency to induce chromosomal AmpC β-lactamases of *P. aeruginosa* and *Enterobacter cloacae* complex, which could otherwise cause resistance development under therapy [14,15,16]. CID has been approved by the Food and Drug Administration (FDA) and the European Medicines Agency (EMA) for the treatment of infections (i.e., complicated urinary-tract infections and hospital-acquired- and ventilator-associated bacterial pneumonia) caused by Gram-negative bacteria in adult patients with limited treatment options [17,18]. 

In vitro data of CID from Germany are scarce. In our previous study published in 2020, CID was found to inhibit 97.2% of a randomly chosen collection of 213 Gram-negative clinical isolates of different species at the investigational susceptibility breakpoint of ≤2 mg/L [19]. The isolates were obtained from patients in intensive care units during a multicentre surveillance study conducted by the Paul-Ehrlich-Society for Infection Therapy (PEG) in 2013. Furthermore, CID was shown to inhibit 88.1% of a collection of 80 carbapenemase-producing clinical isolates from different sources at the investigational susceptibility breakpoint of ≤2 mg/L during that same study [19]. 

The present study aimed (I) to investigate the in vitro activity of CID against Gram-negative pathogens recovered from patients during a more recent multicentre surveillance study conducted by the PEG in 2016/17 and (II) to compare it with the susceptibility against novel β-lactam–β-lactamase-inhibitor (BL-BLI) combinations. 

## 2. Results

### 2.1. Random Sample of Clinical Isolates (Set I) 

MIC distribution data of CID are presented in Table 1. In addition, the MIC values inhibiting 50% and 90% of the isolates (MIC_50_ and MIC_90_) and the number and percent of susceptible and resistant isolates were calculated with the available breakpoints of the European Committee on Antimicrobial Susceptibility Testing (EUCAST) (Table 2). Overall, the CID MIC values ranged from ≤0.03 to ≥64 mg/L, with 98.5% (192/195) of all isolates inhibited if the specifies-specific EUCAST susceptibility breakpoint (Enterobacterales and *P. aeruginosa*) of ≤2 mg/L or the non-species-related pharmacokinetic–pharmacodynamic (PK/PD) breakpoint of ≤ 2 mg/L was applied (*A. baumannii* and *S. maltophilia*).

Among the 111 *Enterobacterales* isolates, CID inhibited 98.2% (109/111) (Table 2). In comparison, 99.1% of the isolates were susceptible to the three comparator agents ceftazidime–avibactam (CTV), imipenem–relebactam (IMR) and meropenem–vaborbactam (MEV), as well as 92.8% to ceftolozane–tazobactam (CTT). The MIC_50_ and MIC_90_ values of CID were 0.12 mg/L and 0.5 mg/L, respectively, as compared to ≤0.06/≤0.06 mg/L for MEV, ≤0.12/0.5 mg/L for CTV, 0.12/0.25 mg/L for IMR and ≤0.25/1 mg/L for CTT. The highest CID MIC (≥64 mg/L) was determined for a respiratory *E. cloacae* complex isolate (PEG-16-51-23) that originated from a nosocomial infection. Sequencing revealed that this isolate encoded the extended-spectrum β-lactamase SHV-12. There were two more isolates detected with CID MICs > 2 mg/L: an NDM-1-encoding *P. aeruginosa* (PEG-16-96-12) and another *E. cloacae* complex (PEG-16-75-70) harbouring *bla*_SHV-12_ (both MICs 4 mg/L). 

Compared to Enterobacterales, MEV was less effective against the 58 *P. aeruginosa* isolates, with 89.7% of the isolates being inhibited at the respective breakpoint (R > 8 mg/L). The susceptibility rates of *P. aeruginosa* against the other agents were comparable to that seen with the Enterobacterales: 98.3% (CID), 96.6% (IMR), 94.8% (CTV) and 91.4% (CTT) (Table 2). Based on the MIC_50/90_ values, CID (0.06/0.5 mg/L) was more active than CTT (1/4 mg/L), CTV (2/8 mg/L), IMR (0.5/2 mg/L) or MEV (1/≥ 16 mg/L). Of the nine *A. baumannii* isolates, all were inhibited by CID at ≤0.5 mg/L. Based on the MIC_50/90_ values, all comparator compounds were less active with MIC_50_ values of 0.5–≥ 16 mg/L and MIC_90_ values of ≥16 mg/L. Two OXA-23-encoding *A. baumannii* isolates (PEG-16-19-60 and PEG-16-22-42) were resistant to IMR (R > 2 mg/L) and exhibited MEV MICs of 16 mg/L. Among the *S. maltophilia* (n = 17) isolates, CID MICs ranged from ≤0.03 to 2 mg/L, with MIC_50/90_ values of 0.06 mg/L and 0.5 mg/L. In comparison, all four comparators were less active in *S. maltophilia* with MIC_50/90_ values of ≥16 mg/L. 

### 2.2. Challenge Organisms (Set II)

The results of set II isolates are displayed together with set I in Table 1 (CID MIC distributions) and Table 2 (MIC_50/90_ and susceptibility/resistance rates, where applicable).

The CID MICs ranged from ≤0.03 to 32 mg/L, with an overall susceptibility rate of 95.3% (101/106) if the specifies-specific EUCAST susceptibility breakpoint (Enterobacterales and *P. aeruginosa*) of ≤2 mg/L or the non-species-related PK/PD breakpoint (*A. baumannii*) of ≤2 mg/L was applied. There were five isolates detected with MICs > 2 mg/L *A. baumannii* (PEG-16-19-65; MIC 32 mg/L), *E. cloacae* complex (PEG-16-41-42, as well as PEG-16-97-14 and PEG-16-97-23; MICs 4 mg/L) and *P. aeruginosa* (PEG-16-14-45; MIC 8 mg/L). Whole-genome sequencing analysis detected the presence of carbapenemase gene *bla*_NDM-1_ in the *A. baumannii* isolate together with disrupted *oprD* and *piuA* genes. The resistant *P. aeruginosa* isolate harboured *bla*_OXA-396_, a variant of the chromosomal class D OXA-50-group in this species, together with the class A β-lactamase PDC-8, and carried a disrupted *oprD*. The three *E. cloacae* complex isolates were only positive for class C ACT-type β-lactamases or CMH-3, which naturally occur in *E. cloacae*.

Among the 53 Enterobacterales isolates, 100% were susceptible to MEV, 98.1% were susceptible to both CTV and IMR and 77.4% were susceptible to CTT. With a susceptibility rate of 94.3% (50/53), CID exhibited slightly lower activity compared to CTV and IMR in Enterobacterales. Overall, the comparator agents were similar or less effective in the 39 *P. aeruginosa* isolates, with susceptibility rates of 61.5% (CTT), 51.3% (CTV), 46.2% (IMR) and 35.9% (MEV), while CID inhibited 97.4% (38/39) of *P. aeruginosa* isolates. Among the CRAB isolates, CID revealed the most potent activity compared to the other compounds with MIC_50/90_ values of 0.12/2 mg/L as opposed to ≥16/≥16 mg/L.

### 2.3. Resistant Isolates (Set I and II)

The CID MIC distributions of ESBL-producing isolates, carbapenemase-producing isolates and colistin-resistant isolates from set I and II are summarized in Table 3, sorted by their respective resistance patterns and mechanisms. ESBL- and carbapenem-resistance determinants are correlated with bacterial species and CID susceptibility in Table 4. 

The CID susceptibility in 47 ESBL-encoding isolates ranged from ≤0.03 mg/L to ≥64 mg/L, with MIC_50/90_ values of 0.5/1 mg/L. There was no difference in CID MIC distribution with regard to different resistance genes, with the exception of two *bla*_NDM-1-_encoding isolates (*A. baumannii* and *P. aeruginosa*) and two SHV-12-encoding *E. cloacae* complex isolates with CID MICs > 2 mg/L. With the exception of CTT, the comparator compounds revealed similar activity against ESBL-producing isolates compared to CID. The MIC_50/90_ values were 0.25/0.5 mg/L for CTV, 0.12/0.25 mg/L for IMR and ≤0.06/0.12 mg/L for MEV. 

The overall CID susceptibility of the 47 ESBL-encoding Enterobacterales isolates was 95.7% (45/47). CTV, IMR and MEV were able to inhibit 100% of the isolates at their respective breakpoints, with only the CTT susceptibility being lower at 85.1%. 

Among the 30 carbapenemase-producing isolates, CID MICs ranged from ≤0.03 to 32 mg/L, with 93.3% (28/30) of the isolates being inhibited at a concentration of ≤2 mg/L. The MIC_50/90_ values were 0.5/2 mg/L. Fourteen of fifteen carbapenemase-producing *A. baumannii* isolates (93.3%) were inhibited at a CID concentration ≤ 2 mg/L. Regarding carbapenemase-producing *P. aeruginosa* isolates, 91.7% (11/12) exhibited CID susceptibility. CID-resistant isolates of both species (*A. baumannii* PEG-16-19-65 and *P. aeruginosa* PEG-16-96-12) encoded *bla*_NDM-1_ and revealed MIC values of the comparator substances of 16 mg/L each. With the exception of one *A. baumannii* isolate, all carbapenemase-encoding isolates were susceptible to colistin. The colistin-resistant isolate (MIC 8 mg/L) was susceptible to CID but resistant to all of the other tested compounds. Overall, the comparators revealed reduced activity in carbapenemase-encoding isolates compared to CID with MIC_50/90_ values greater or equal to their highest concentration tested. 

In colistin-resistant isolates (n = 37), CID MICs ranged from ≤ 0.03 to 4 mg/L, with MIC_50/90_ values of 0.25/1 mg/L. Based on MIC_50/90_ values, CID activity was comparable to CTT (0.5/2 mg/L), CTV (0.5/4 mg/L) and IMR (0.25/1 mg/L). The distribution of MEV MICs revealed a lower MIC_50_ value compared to the other compounds (MIC_50/90_s: ≤0.06/2 mg/L). All colistin-resistant *P. aeruginosa* isolates (n = 14) were inhibited by CID and the comparator substances. The two colistin-resistant *A. baumannii* isolates (PEG-16-36-64 and PEG-16-50-50) were inhibited by a CID concentration < 2 mg/L but revealed different MIC results with regard to the comparator substances. PEG-16-36-64 was susceptible against all the other substances (MIC range 0.25–0.5 mg/L), while the *bla*_OXA-23-like-_positive PEG-16-50-50 exhibited MIC values of 16 mg/L against CTT, CZA, IMR and MEV.

## 3. Discussion

The WHO has designated antimicrobial resistance as one of the top ten global public health threats. Of great concern are carbapenem-resistant non-fermenting Gram-negative bacteria such as *A. baumannii* and *P. aeruginosa*, as well as Enterobacterales species with acquired resistance against carbapenems or third-generation cephalosporins, all possessing a high risk of severely limited treatment options. In contrast to resistance against third-generation cephalosporins, carbapenem resistance is still rarely encountered in Enterobacterales species such as *E. coli* and *K. pneumoniae* in Germany. For example, the surveillance study originated by the Paul-Ehrlich-Society for Infection Therapy in 2016/17 revealed resistance rates of 0% against imipenem and meropenem in 571 *E. coli* isolates and 1.6% (5/318) against both substances in *K. pneumoniae* isolates. According to the annual surveillance data for Germany reported to the European Centre for Disease Prevention and Control (ECDC), the rates of carbapenem-resistant *E. coli* and *K. pneumoniae* isolates were 0.0% and 0.8% in 2021 (out of a total of 29,105 and 6538 isolates tested, respectively) compared to resistance rates against third-generation cephalosporins of 9.1% (2641/29,021) and 10.4% (678/6538) (Surveillance Atlas of Infectious Diseases (europa.eu); data source: invasive isolates). In *Acinetobacter* spp. and *P. aeruginosa*, carbapenem-resistance was more frequently detected with 4.3% (26/605) and 14.8% (425/2864), respectively.

The antimicrobial agents compared in this study are considered promising compounds in the treatment of infections with the above-mentioned organisms when no other options are available. In contrast to the comparators, CID possesses activity against a variety of Gram-negative species and β-lactamases of all Ambler classes, including OXA-encoding *A. baumannii*, MBL-producing organisms and *S. maltophilia*, which is intrinsically resistant against multiple antimicrobial agents, including carbapenems [20]. Unlike the other compounds, MEV is not available in Germany yet.

In the current study, CID showed broad activity with overall inhibition rates of 98.5% (set I, MIC_50/90_s 0.12/0.5 mg/L) and 95.3% (set II, MIC_50/90_s 0.5/1 mg/L) at ≤2 mg/L (Table 1), which was in accordance with our previous study [19]. The other compounds showed comparable potent activity in Enterobacterales (susceptibility rates >92%), with reduced activity of CTT in challenge isolates (set II) (Table 2). This result might have been expected due to the compound-specific spectrum. In *P. aeruginosa*, MEV displayed reduced activity in set I isolates compared to CID (Table 2). In set II isolates of *P. aeruginosa*, all comparators showed decreased activity, with inhibition rates ranging from 36% to 62%. In accordance with our study, the SENTRY surveillance study reported similar CID MIC_50/90_ values based on larger strain collections of 8047 Enterobacterales and 2282 *P. aeruginosa* isolates from Europe and the United States with 0.06/0.5 mg/L in Enterobacterales and 0.12/0.5 mg/L in *P. aeruginosa* [21]. 

Our study observed more potent CID activity against *A. baumannii* and *S. maltophilia* isolates than the comparators (Table 2). Similar results were reported previously in different studies investigating bacterial isolates from the United States and from Europe [21,22,23]. In the current study, 100% of *S. maltophilia* and *A. baumannii* random isolates (set I) were inhibited by CID at a concentration of ≤2 mg/L, while there was only one isolate detected among fourteen CRAB isolates (set II) with an MIC of >2 mg/L. However, due to the small sample size of *S. maltophilia* and *A. baumannii*, these data should be considered with caution. Naas et al. investigated a total number of 103 *S. maltophilia* and 161 *A. baumanni* isolates and reported comparable CID activity against both species and almost identical MIC_50/90_ values of 0.06 mg/L and 0.25 mg/L [22]. Karlowsky et al. analysed data of five annual SIDERO-WT surveillance studies from 2014 to 2019 based on >47,000 bacterial isolates, including 2030 *S. maltophilia* [24]. They detected a lower CID susceptibility rate compared to our study of 98.6% with a wider MIC range (≤0.004–8 mg/L) but similar MIC_50/90_ values of 0.06/0.25 mg/L. However, studies on CRAB yielded conflicting results. A recent study by Mushtaq et al. investigated 99 *A. baumannii* isolates encoding various OXA-β-lactamases, with the majority being OXA-23 (n = 41), as well as NDM enzymes (n = 20) [25]. In this study, CID at 2 mg/L was only able to inhibit 80.8% of the investigated isolates. In contrast, Delgado-Valverde et al. only reported reduced CID efficacy in OXA-24/40 expressing *A. baumannii* (n = 25), while isolates harbouring OXA-58 or OXA-23 were all susceptible (n = 75) [23]. 

Overall, our study revealed the broad activity of CID against ESBL- and carbapenemase-producing isolates, as well as colistin-resistant isolates, with the majority of isolates inhibited at ≤2 mg/L (Table 3). Isolates with CID MIC values > 2 mg/L harboured acquired β-lactamases such as NDM-1-like (*A. baumannii* and *P. aeruginosa*) and SHV-12 (*E. cloacae* complex), or the naturally occurring β-lactamases such as class C ACT-type, CMH-3 (both *E. cloacae*), as well as PDC-8 together with class D OXA-396 (*P. aeruginosa*) (Table 4). The correlation of NDM production with CID non-susceptibility has been observed previously [10,25]. In accordance, some studies showed that the cloning of *bla*_NDM-1_ in *E. coli* resulted in an increase in the CID MIC from 0.5 mg/L to 4 mg/L, while the cloning of other β-lactamase genes such as *bla*_ACT-_type or *bla*_OXA-23_ revealed lower MICs of 0.125 to 0.5 mg/L [26,27]. In our study, *bla*_NDM-1_-carrying isolates also revealed CID MICs > 2 mg/L, but as only two isolates were included, our data are of limited value in further support of the association between the presence of NDM-1 and reduced CID susceptibility. Of note, the CID-resistant *A. baumannii* isolate PEG-16-19-65 revealed the disruption of the *piuA* gene, which encodes a siderophore receptor that might be needed for efficient CID uptake, as has been shown previously for its homologue in *P. aeruginosa* [28,29]. Furthermore, the disruption of *oprD* porin genes was detected in PEG-16-19-65 and the CID-resistant *P. aeruginosa* PEG-16-14-45. However, an association between CID resistance and disrupted *oprD* genes has not been described yet.

In conclusion, CID revealed potent activity against Enterobacterales and non-fermenting Gram-negative bacterial isolates from Germany. The association of CID non-susceptibility with a particular resistance determinant seemed to be unlikely, while the presence of *bla*_NDM-1_ might be an exception to this and requires further investigation. CID activity was superior to the comparators against *A. baumannii*, *S. maltophilia* and challenge isolates of *P. aeruginosa*. In addition to ESBL-producing isolates, the majority of CP-producing and colistin-resistant isolates were inhibited at a CID concentration ≤2 mg/L, indicating good activity of CID in clinically relevant organisms. 

## 4. Materials and Methods

### 4.1. Bacterial Isolates 

In total, 301 Gram-negative bacterial isolates were investigated in this study. All isolates were obtained from patient samples collected at 22 German microbiological laboratories during a multicentre surveillance study conducted by the PEG in 2016/17. The majority of laboratories were affiliated with tertiary-care medical centres. Two sets of isolates were selected: random samples (set I) and challenge organisms (set II).

#### 4.1.1. Random Sample of Clinical Isolates (Set I)

Set I included 195 isolates (selected from a total number of 511 isolates) which encompassed 111 Enterobacterales and 84 non-fermenting bacteria which were obtained from the respiratory tract (n = 117) and from blood culture (n = 78). Isolates were randomly selected; routine clinical isolates, also including ESBL- and carbapenemase-producing isolates; as well as colistin-resistant isolates. The Enterobacterales included *E. coli* (n = 52), *K. pneumoniae* (n = 34) and *E. cloacae* complex (n = 25) isolates. The non-fermenting bacteria included *A. baumannii* (n = 9)*, P. aeruginosa* (n = 58) and *S. maltophilia* (n = 17) isolates. 

Among the Enterobacterales, ten *E. coli* and five *K. pneumoniae* isolates were ESBL producers, which were investigated via PCR/Sanger sequencing or whole-genome sequencing in different reference laboratories as part of the PEG study. All of the ESBL *K. pneumoniae* isolates as well as five ESBL *E. coli* isolates carried the *bla*_CTX-M-15_ gene. Additionally, three of these *K. pneumoniae* isolates also encoded SHV-40 (n = 1) or SHV-28 (n = 2). The remaining ESBL *E. coli* isolates carried *bla*_CTX-M-1_ (n = 4) or *bla*_CTX-M-27_ (n = 1). Few isolates of set I encoded carbapenemases: *K. pneumoniae* (n = 1) encoded VIM-1, *A. baumannii* (n = 2) encoded OXA-23 and *P. aeruginosa* (n = 1) encoded NDM-1. Seven Isolates were colistin-resistant (*E. cloacae* (n = 3), *E. coli* (n = 1) and *K. pneumoniae* (n = 3)).

#### 4.1.2. Challenge Organisms (Set II)

Set II comprised 53 Enterobacterales and 53 non-fermenting bacteria with either confirmed carbapenemases or ESBL genes or with phenotypic colistin resistance or „meropenem non-susceptibility”. „Meropenem-non-susceptible” isolates included *A. baumannii* and *P. aeruginosa* isolates with a meropenem MIC above 8 mg/L and Enterobacterales that were either ertapenem-resistant (R > 0.5 mg/L) and/or possessed a meropenem MIC of >0.12 mg/L (meropenem screening cut-off value according to EUCAST). In contrast to set I, isolates of set II were especially enriched for ESBL- and carbapenemase-producing isolates as well as colistin-resistant isolates. The investigation of resistance genes was part of the multicentre study conducted by the PEG and was performed via PCR/Sanger sequencing or whole-genome sequencing in different reference laboratories [30,31]. *A. baumannii* isolates of this set encoded the following β-lactamases: NDM-1 (n = 1), OXA-58 (n = 1), or OXA-23 (n = 11). The *E. coli* isolates of this set produced CTX-M-1 (n = 3), CTX-M-14 (n = 2), CTX-M-15 (n = 9), CTX-M-15 plus CTX-M-27 (n = 1), CTX-M-27 (n = 3), or CTX-M-55 (n = 1). *P. aeruginosa* isolates of this set encoded the following β-lactamases: GIM-1 (n = 2), IMP-7 (n = 2), IMP-13 (n = 1), VIM-1 (n = 2), VIM-2 (n = 3) and VIM-5 (n = 1). *K. pneumoniae* isolates of this set encoded the following β-lactamases: CTX-M-3 (n = 1), CTX-M-15 (n = 10) and VIM-1 (n = 1). One *E. cloacae* complex isolate encoded OXA-48.

### 4.2. Species Identification 

The verification of species identification was performed via matrix-assisted laser desorption ionization–time of flight (MALDI-TOF) mass spectrometry (MALDI Biotyper, Microflex, Bruker Daltonics GmbH, Bremen, Germany).

### 4.3. Antimicrobial Susceptibility Testing

The following antimicrobial agents were tested with the noted ranges: ceftazidime–avibactam (CTV) (0.12/4–8/4 mg/L), ceftolozane–tazobactam (CTT) (0.25/4–8/4 mg/L), imipenem–relebactam (IMR) (0.03/4–8/4 mg/L), meropenem–vaborbactam (MEV) (0.25/8–8/8 mg/L) and cefiderocol (0.06–32 mg/L). MICs were determined using the broth microdilution procedure with geometric twofold serial dilutions according to the international standard ISO 20776-1 [32]. The susceptibilities of the comparator compounds were analysed using industrially manufactured, ready-to-use 96-well plates and cation-adjusted Mueller Hinton broth (CAMHB) from ThermoFisher (Waltham, MA, USA). In parallel, cefiderocol (CID) MICs were determined using freshly prepared in-house plates and iron-depleted CAMHB (ID-CAMHB) provided by the International Health Management Associates Inc. (IHMA, Schaumburg, IL, USA). The final test volume was 100 µL per well. The final bacterial inoculum was approximately 5 × 10^5^ CFU/mL (range 2–8 × 10^5^). Panels were incubated at 35 ± 1°C for 18 ± 2 h. The MICs were read visually and, as far as possible, interpreted according to the species-specific clinical breakpoints approved by the EUCAST (version 13.0, January 2023: S (susceptible; standard dosing regimen), I (susceptible; increased exposure) and R (resistant) [33]. For CID, the species-specific clinical breakpoints of CID for Enterobacterales and *P. aeruginosa* of ≤2 mg/L (S) and >2 mg/L (R) were applied. For other species, the EUCAST-approved pharmacokinetic–pharmacodynamic (PK-PD; non-species related) breakpoints were applied (≤2 mg/L (S) and >2 mg/L (R)). Reference strains *E. coli* ATCC 25922 and *P. aeruginosa* ATCC 27853 were used for quality control. 

### 4.4. Molecular Analysis of CID-Resistant Isolates

Isolates with CID MICs > 2 mg/L were sent to the International Health Management Associates (IHMA) for whole-genome sequencing.

### 4.5. Statistical Evaluation

The statistical significance of differences in susceptibility rates was judged by comparing 95% confidence intervals (CIs). Intervals were constructed using the Newcombe–Wilson method without continuity correction. If no rate was contained in the CI of the other one, significance of *p* < 0.05 was assumed. 

## Figures and Tables

**Table 1 antibiotics-12-00864-t001:** In vitro activity of cefiderocol against Gram-negative pathogens.

Species	n	MIC (mg/L)											
		≤0.03	0.06	0.12	0.25	0.5	1	2	4	8	16	32	≥64
**Random sample of isolates (set I, n = 195)**													
*E. coli*	52	14	12	10	5	9	2						
*K. pneumoniae*	34	10	7	6	3	7		1					
*E. cloacae* complex	25	1	1	3	2	13	3		1				1
*P. aeruginosa*	58	4	25	17	1	6	3	1	1				
*A. baumannii*	9		5	1		3							
*S. maltophilia*	17	2	9	3		2		1					
Subtotal	195	31	59	40	11	40	8	3	2				1
**Sample of resistant isolates (set II, n = 106) ^1^**													
*E. coli*	22	2	3	2		11	4						
*K. pneumoniae*	15	1	1	4	2	4	3						
*E. cloacae* complex	16			2	1	9	1		3				
*P. aeruginosa*	39	2	7	6	5	9	9	1		1			
*A. baumannii*	14	6	1	1	4		1				1		
Subtotal	106	6	22	15	9	41	17	2	3	1		1	
**Total**	**301**	**36**	**76**	**55**	**20**	**77**	**25**	**5**	**5**	**1**		**1**	**1**

Abbreviations: n, number of isolates; MIC, minimum inhibitory concentration. ^1^ Set II comprised ESBL producers, possible carbapenemase producers and/or colistin-resistant isolates.

**Table 2 antibiotics-12-00864-t002:** In vitro activity of cefiderocol and four β-lactam–β-lactamase inhibitor combinations against Gram-negative pathogens by set of isolates (n = 301).

Random Sample of Isolates (Set I, n = 195)					Sample of Resistant Isolates (Set II, n = 106) ^1^				
Antibacterial Agent	MIC_50_ (mg/L)	MIC_90_ (mg/L)	Number (%) of Isolates		Antibacterial Agent	MIC_50_ (mg/L)	MIC_90_ (mg/L)	Number (%) of Isolates	
			S	R				S	R
**Enterobacterales (n = 111) ^2^**					**Enterobacterales (n = 53) ^3^**				
CID	0.12	0.5	109 (98.2)	2 (1.8)	CID	0.5	1	50 (94.3)	3 (5.7)
CTT	≤0.25	1	103 (92.8)	8 (7.2)	CTT	0.5	≥16	41 (77.4)	12 (22.6)
CTV	≤0.12	0.5	110 (99.1)	1 (0.9)	CTV	0.25	1	52 (98.1)	1 (1.9)
IMR	0.12	0.25	110 (99.1)	1 (0.9)	IMR	0.12	0.5	52 (98.1)	1 (1.9)
MEV	≤0.06	≤0.06	110 (99.1)	1 (0.9)	MEV	≤0.06	0.12	53 (100)	0 (0)
***P. aeruginosa* (n = 58)**					***P. aeruginosa* (n = 39)**				
CID	0.06	0.5	57 (98.3)	1 (1.7)	CID	0.5	1	38 (97.4)	1 (2.6)
CTT	1	4	53 (91.4)	5 (8.6)	CTT	2	≥16	24 (61.5)	15 (38.5)
CTV	2	8	55 (94.8)	3 (5.2)	CTV	8	≥16	20 (51.3)	19 (48.7)
IMR	0.5	2	56 (96.6)	2 (3.4)	IMR	4	≥16	18 (46.2)	21 (53.8)
MEV	1	≥16	52 (89.7)	6 (10.3)	MEV	≥16	≥16	14 (35.9)	25 (64.1)
***A. baumannii* (n = 9)**					***A. baumannii* (n = 14)**				
CID	0.06	0.5	No EUCAST breakpoints		CID	0.12	2	No EUCAST breakpoints	
CTT	2	≥16	No EUCAST breakpoints		CTT	≥16	≥16	No EUCAST breakpoints	
CTV	≥16	≥16	No EUCAST breakpoints		CTV	≥16	≥16	No EUCAST breakpoints	
IMR	0.5	≥16	7 (77.8)	2 (22.2)	IMR	≥16	≥16	1 (7.1)	13 (92.9)
MEV	0.5	≥16	No EUCAST breakpoints		MEV	≥16	≥16	No EUCAST breakpoints	
***S. maltophilia* (n = 17)**									
CID	0.06	0.5	No EUCAST breakpoints						
CTT	≥16	≥16	No EUCAST breakpoints						
CTV	≥16	≥16	No EUCAST breakpoints						
IMR	≥16	≥16	No EUCAST breakpoints						
MEV	≥16	≥16	No EUCAST breakpoints						

^1^ See footnotes of Table 1 for details. ^2^
*Enterobacter cloacae* complex (n = 25), *Escherichia coli* (n = 52), *Klebsiella pneumoniae* (n = 34). ^3^
*Enterobacter cloacae* complex (n = 16), *Escherichia coli* (n = 22), *Klebsiella pneumoniae* (n = 15). Abbreviations: S, susceptible; R, resistant; CID, cefiderocol; CTT, ceftolozane–tazobactam; CTV, ceftazidime–avibactam; IMR, imipenem–relebactam; MEV, meropenem–vaborbactam.

**Table 3 antibiotics-12-00864-t003:** In vitro activity of cefiderocol against various subgroups of Gram-negative isolates (set I and set II).

Bacterial Group	MIC (mg/L)											
	≤0.03	0.06	0.12	0.25	0.5	1	2	4	8	16	32	≥64
**ESBL-producing Enterobacterales (n = 47) ^1^**												
CID	**3**	4	6	4	21	7		1				1
CTT				**2**	25	9	4		2	*5*		
CTV			**14**	20	9	3	1					
IMR			32	11	2	1	1					
MEV		**42**	3				1	1				
**Carbapenemase-producing isolates (n = 30) ^2^**												
CID	**1**	6	1	2	12	4	2	1			1	
CTT								2	1	*27*		
CTV						1			1	*28*		
IMR						1			2	*27*		
MEV							1	1	1	*27*		
**Colistin-resistant isolates (n = 37) ^3^**												
CID	**1**	86	9	4	11	2	1	1				
CTT				**13**	12	8	1	1	1	*1*		
CTV			**8**	8	4	5	8	2	1	*1*		
IMR			9	12	10	3	2			*1*		
MEV		**22**		2	2	7	1		2	*1*		

^1^ *E. coli* (n = 29), *K. pneumoniae* (n = 16), *E. cloacae* complex (n = 2), ^2^ *A. baumannii* (n = 15), E. cloacae complex (n = 1), *K. pneumoniae* (n = 2), *P. aeruginosa* (n = 12), ^3^ *A. baumannii* (n = 2), E. cloacae complex (n = 13), *E. coli* (n = 4), *K. pneumoniae* (n = 4), *P. aeruginosa* (n = 14). Abbreviations: CID, cefiderocol; CTT, ceftolozane–tazobactam; CTV, ceftazidime–avibactam; IMR, imipenem–relebactam; MEV, meropenem–vaborbactam. Numbers in bold include isolates with MIC < value shown; numbers in italics include isolates with MIC > the highest concentration tested.

**Table 4 antibiotics-12-00864-t004:** Distribution of cefiderocol MICs in isolates with characterized ESBL/carbapenemase genes or colistin-resistant isolates.

**Species**	**CTX-M-Group**	**ESBL Type**	**MIC (mg/L)**											
			**≤0.03**	**0.06**	**0.12**	**0.25**	**0.5**	**1**	**2**	**4**	**8**	**16**	**32**	**≥64**
*E. cloacae* complex	-	SHV-12 (n = 2)								1				1
*E. coli*	1/2	CTX-M-1 (n = 7)	2	3	2									
		CTX-M-15 (n = 14)			1	1	9	3						
		CTX-M-55 (n = 1)					1							
	9	CTX-M-14 (n = 2)		1				1						
		CTX-M-27 (n = 4)					4							
	1/2 + 9	CTX-M-15 + CTX-M-27 (n = 1)						1						
	**Total (n = 29)**		**2**	**4**	**3**	**1**	**14**	**5**						
**Species**	**CTX-M-Group**	**ESBL Type**	**MIC (mg/L)**											
			**≤0.03**	**0.06**	**0.12**	**0.25**	**0.5**	**1**	**2**	**4**	**8**	**16**	**32**	**≥64**
*K. pneumoniae*	1/2	CTX-M-3 + SHV-11 (n = 1)				1								
		CTX-M-15 (n = 4)			1	1	1	1						
		CTX-M-15 + SHV-11 (n = 4)	1		1		1	1						
		CTX-M-15 + SHV-28 (n = 4)			1		3							
		CTX-M-15 + SHV-40 (n = 1)					1							
		CTX-M-15 + SHV-76 (n = 1)					1							
		CTX-M-15 + SHV-201 (n = 1)				1								
	**Total (n = 16)**		**1**		**3**	**3**	**7**	**2**						
	**Ambler Class**	**Type of Carbapenemase**	**MIC (mg/L)**											
			**≤0.03**	**0.06**	**0.12**	**0.25**	**0.5**	**1**	**2**	**4**	**8**	**16**	**32**	**≥64**
*A. baumannii*	B	NDM-1 (n = 1)											1	
	D	OXA-23 (n = 13)		4	1	1	6		1					
		OXA-58 (n = 1)		1										
*E. cloacae* complex	D	OXA-48 (n = 1)					1							
*K. pneumoniae*	B	VIM-1 (n = 2)				1	1							
*P. aeruginosa*	B	GIM-1 (n = 2)	1	1										
		IMP-7 (n = 2)						1	1					
		IMP-13 (n = 1)					1							
		NDM-1 (n = 1)								1				
		VIM-1 (n = 2)					1	1						
		VIM-2 (n = 3)					2	1						
		VIM-5 (n = 1)						1						
	**Total (n = 30)**		**1**	**6**	**1**	**2**	**12**	**4**	**2**	**1**			**1**	

## Data Availability

Not applicable.

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
