# Peer review of "In Vitro Activity of Cefiderocol against Clinical Gram-Negative Isolates Originating from Germany in 2016/17"

_antibiotics, 2023, doi:10.3390/antibiotics12050864_

Round 1

Reviewer 1 Report

General comments

This is a very illustrative work on the current resistance situation of gram-negative bacilli in Germany.

It analyzes the sensitivity of two groups of bacteria, the second with the highest degree of antibiotic resistance, to cefiderocol and some modern combinations of beta-lactams with beta-lactamase inhibitors.

The work has great merit because, fortunately, it is difficult to find carbapenem-resistant strains in Germany. The study demonstrates very well the greater clinical usefulness of Cefiderocol against Acinetobacter baumannii, Stenotrophomonas maltophilia and, to a lesser extent, carbapenem-resistant enterobacteria.

Specific comments

Table 1. Second column must be widened to adequately display the numbers contained in it.

Author Response

Thank you very much for your comments.

The table was wider in the manuscript version we have submitted. I guess it has been compressed due to formatting purposes. To address your comment we have decreased the font size and were able to adapt the width of the columns accordingly.

Reviewer 2 Report

In-vitro activity of cefiderocol against clinical Gram-negative isolates originating from Germany in 2016/17

Overall comments:

The study was well devised and of clinical relevance. The authors have evaluated the next generation siderophore cephalosporin cefiderocol. They tested the susceptibility of 301 clinical isolates collected between 2016 and 2017. These isolates included both random and resistant isolates from gram negative species such as P. aeruginosa, A.baumannii, K.pneumoniae, S. maltophilia ,E.coli and E.cloacae. Cefiderocol performed very well with over 97% of the total isolates testing being susceptible to cefiderocol. The 8 isolates that exhibited resistance were found to contain beta lactamase genes such as (bla) genes blaNDM-1, blaSHV-12 and naturally occurring blaOXA-396, blaACT-type and blaCMH-3.

Specific comments:

1) Have the authors seen any literature or information regarding the ability of the tested comparators beta-lactamase inhibitors such as vaborbactam or relebactam being able to overcome any of the genes found in cefiderocol resistance such as blaNDM-1.

2) Were any of the cefiderocol resistant isolates susceptible to any of the compared combinations? The Enterobacterales isolates had less resistant isolates against CTV, IMR and MEV. If possible the authors should look into testing one or two of these beta lactamase inhibitors (V or R) with cefiderocol and seeing if they have synergistic or additive effects against the few cefiderocol resistant isolates identified. While cefiderocol displayed exceptional potency, resistance to this new drug will grow overtime and having potential combinations which can later be used to be of great clinical value.

Author Response

Thank you very much for your comments.

Here are my answers:

1) There is no activity of vaborbactam or relebactam against the Ambler class B NDM-1 metallo-beta-lactamase.

2)  Most of the cefiderocol-resistant isolates were susceptible to Meropenem and Meropenem/Vaborbactam, as well as Imipenem and Imipenem/Relebactam. However, the NDM-1-encoding isolates were resistant against both carbapenems and their combinations with the respective beta-lactamase inhibitor, which supports the statement that the inhibitors are not active against this type of beta-lactamase. Taken this into account as well as the fact that retesting of even a few isolates will take weeks because we would have to order the beta-lactamase inhibitors and prepare in-house plates etc. we have to reject the request.

Reviewer 3 Report

Wohlfarth et al reported in vitro activity of cefiderocol against clinical Gram-negative  isolates originating from Germany in 2016/17.  Overall, the manuscript was well organized and their findings are desirable for clinicians and researchers around the world. However, there are some minor concerns about this study that warrant consideration prior to publication. Please see my comments below.

1. For Line 44, the “,” after “and” is not necessary. Moreover, adding a few references to support the claim of the first sentence is suggested.

2.  For Table 1, the column of “n” need a better format to show the numbers in the same row. It may be better to let audience know what “n” refers to.

3. For Tables 3, the footnotes for the full names of antibiotics used are missing and the second row needs a modification for the formatting to make sure the numbers are on the same row.

5. Some of the bacterial species were not using italics, some are in the key words session and some are in the body content.

6. “In vitro” is not consistent, some are in italics, some are not, and some have a “-” in between the two words.

7. For those isolates went through whole genome sequencing, it would be better to add accession numbers or let audience know the data-sharing plan.

8. For references, some are with clickable doi link, some are not, I would suggest to be consistent and to match the journal’s requirement.

Minor editing of English language required.

Author Response

Thank you very much for your comments.

My answers are as follows:

1) We deleted the comma and added two references.

2) We decreased the font size and were able to adapt the table as requested.

3) We decreased the font size and were able to adapt the table as requested.

5) Adapted

6) Adapted

7) We have asked for the accession numbers but are still waiting for the response.

8) Adapted